# Controls on Palaeogene deep-sea diatom-bearing sediment deposition and comparison with shallow marine environments

Cécile Figus[1,2], Johan Renaudie[3], Or M. Bialik[4,5], and Jakub Witkowski[1]

[1]Institute of Marine and Environmental Sciences, University of Szczecin, 70-383 Szczecin, Poland
[2]Doctoral School, University of Szczecin, 70-383 Szczecin, Poland
[3]FB1 Dynamik der Natur, Museum für Naturkunde, 10115 Berlin, Germany
[4]Institute of Geology and Palaeontology, University of Münster, 48149 Münster, Germany
[5]Dr. Moses Strauss Department of Marine Geosciences, The Leon H. Charney School of Marine Sciences, University of Haifa, Mount Carmel, 31905 Haifa, Israel

**Correspondence:** Cécile Figus (cecile.figus@phd.usz.edu.pl)

**Abstract.** Diatoms are the key players in the present-day global biogeochemical cycles. Yet, the diatom flux response to the dynamically changing climates of the Palaeogene has long been a subject to divergent interpretations. We present a synthesis of Palaeogene deep-sea diatom-bearing sediment occurrences in time and space, in order to gain new insight into inter-basin and latitudinal distribution of diatom accumulation zones from the Cretaceous/Palaeogene boundary to the Oligocene/Miocene

transition. Our dataset includes 189 sites drilled in the Atlantic, Pacific, and Indian oceans, and in the Arctic. It suggests that the number and distribution of deep-sea diatom-bearing sediment occurrences is mainly controlled by the nutrient availability and ocean circulation. Climate appears to have only an indirect correlation with our results, which may be linked to the expansion of the Antarctic Ice Sheet during the Oligocene global cooling. A comparison of our results with the temporal distribution of shallow marine diatomite occurrences (Figus et al., 2024a) suggests that the increase in the number of deep-sea diatom-bearing

sediment occurrences (particularly in the Atlantic) during the diatomite gap (~46 to ~44 Ma) in shallow marine settings is indirectly related to the tectonic reorganizations occurring during this period: palaeogeographic changes caused the cessation of shallow marine diatomite deposition, and increased nutrient availability in the oceans through continental weathering intensification. We also challenge the previous views on geographic shifts in the key loci of biogenic silica accumulation, which generally indicate that as global cooling progressed through middle and late Eocene, the Southern Ocean was gradually be-

coming the key biogenic silica sink. Our synthesis shows – albeit in a non-quantitative perspective – that through most of the Palaeogene, low to mid-latitude areas, especially in the Atlantic Ocean, were the locus of widespread biogenic silica deposition and burial.

## 1 Introduction

The distribution of diatoms in the modern oceans is closely related to the cycling of carbon and silicon, and to the supply of

nutrients: nitrogen, phosphorus and iron are particularly important for diatoms, acting as limiting nutrients for phytoplankton proliferation (Hutchins and Bruland, 1998; Tyrrell, 1999). Within the biological pump, diatoms are responsible for a significant

proportion of the export of organic carbon through photosynthesis. Their interactions with the silicon cycle also make them the main primary producer and exporter of organic silica in the oceans (Renaudie, 2016). They use dissolved silica ($H_4SiO_4$), mostly supplied to the oceans by riverine input or recycled directly into the water column, to grow their siliceous frustules

(Tréguer et al., 2021). The supply of $H_4SiO_4$ is closely linked to the weathering of continental rocks, which can increase as a result of an intensification of the hydrological cycle, tectonic activity or an increase in atmospheric $CO_2$, for example (Berner et al., 1983; Penman et al., 2019). In surface waters, and particularly in conditions favourable for blooming, such as upwelling zones, diatoms are more likely to be grazed, while bacterial activity plays a major role in the dissolution of frustules in warm water areas, enhancing the recycling of dissolved silica (Ragueneau et al., 2006). Another aspect of the biological pump is

represented by the processes accelerating the diatom sinking in the water column and the consequent burial of diatom frustules. Among them, the massive settling of large aggregates of bloom-forming or stratified-adapted diatoms ("self-sedimentation" and 'fall dump", respectively) play a prominent role in enhancing diatom-bearing sediment formation (Smetacek, 1985; Grimm et al., 1997; Kemp et al., 2000). As a result, different types of sediments (e.g. diatom oozes, diatomites, cherts, etc.) are formed, depending on their diatom content (Zahajská et al., 2020).

In today's oceans, diatoms appear to be distributed mainly in nutrient-rich cold waters, such as in high latitude regions and upwelling zones (Tréguer et al., 2018). However, it is difficult to reconstruct their geographic distribution in the past. The Palaeogene (66 to 23 million years ago [Ma]) was a period characterized by an initial warm climate, gradually cooling after the termination of the Early Eocene Climate Optimum (EECO; ~53 to ~49 Ma). Despite several studies examining temporal trends in early Cenozoic diatom-bearing sediments from various locations (Barron et al., 2015; Wade et al., 2020; Witkowski

et al., 2021), the global distribution of deep-sea diatom-bearing sediments in the Palaeogene and the involvement of diatoms in biogeochemical cycling in the Palaeogene oceans tend to be based on very limited datasets (Bryłka et al., 2024). In addition, the mechanisms impacting diatom deposition during this period remain poorly understood. In order to better understand which and how global mechanisms play a role in diatom productivity and in the distribution of diatom-bearing sediments, it is necessary to synthesize data on a global scale.

A previous synthesis of data was created by Figus et al. (2024a), to determine the global distribution of diatomite occurrences iIn shallow marine environments, and which factors impacted diatomite deposition throughout the Palaeogene. Their results suggest the importance of climatic and tectonic controls on diatomite accumulation, with a stronger influence of ocean circulation in open ocean settings from ~43.5 Ma. Furthermore, they interpret a gap in diatomite deposition in epicontinental seas between ~46 and ~44 Ma, implying a contrasting pattern compared to the western North Atlantic, where biosiliceous

fluxes peak between ~46 and ~42 Ma (Witkowski et al., 2021). Comparing these results raises a number of questions, including whether the Pacific and Indian oceans behaved similarly to the Atlantic during this period and, if not, why they reacted differently. Furthermore, is this apparent inverse relationship between the Atlantic and the shallow marine environments the result of a causal link, the influence of a common factor or an indirect correlation?

        To address these questions, we present a global compilation of Palaeogene deep-sea diatom-bearing sediment occurrences,

based on data reports from the Deep Sea Drilling Project (DSDP, 1966 to 1983), the Ocean Drilling Program (ODP, 1983 to 2003), the Integrated Ocean Drilling Program (IODP, 2003 to 2013) and the International Ocean Discovery Program (IODP,

2013 to 2024). Since 1966, over 400 expeditions have been carried out, enabling the collection of sediments from the Mesozoic to the present day, on nearly a global scale. However, these data have never been synthesized to examine the global repartition of Palaeogene diatom-bearing sediments. The well-documented reports of these drilling campaigns provide a robust foundation for the creation of our compilation, allowing its comparison with the dataset in Figus et al. (2024a) and various proxies. Based on this data analysis, we examine for the first time the importance of climate, tectonics and ocean circulation for the global deposition of deep-sea diatom-bearing sediments during the Palaeogene.

## 2 Materials and methods

### 2.1 Lithology and terminology

Fossil marine diatoms are present in shallow and deep marine environments, where they can form (or be included in) different types of biogenic sediments. Mazzullo et al. (1988) denoted that the component of a sediment must represent at least 10% of its composition to be regarded as a modifier. Zahajská et al. (2020) propose the following classification: diatoms present in $> 10\%$ of a sediment can form a diatomaceous sediment (between 10 and 50% diatoms), a diatomaceous ooze ($> 50\%$), a diatomaceous sedimentary rock (lithified diatomaceous sediment), or a diatomite (consolidated diatomaceous ooze). At this stage, the silica contained in diatoms is known as opal-A (i.e., amorphous form). However, diagenetic processes, intensifying with burial depth (and therefore with temperature and pressure), modify the structures of these sediments into shale (10 to 50% siliceous microfossils), porcelanite (20 to 85%) and chert ($> 80\%$), where the opal-A becomes a silica polymorph called opal-CT (Rice et al., 1995; Yanchilina et al., 2020). Since diagenesis modifies the structure of the sediment, determining the biosiliceous origin of these rocks can be difficult, and the literature is often imprecise on the presence of diatoms, radiolarians or sponge spicules in cherts (Witkowski et al., 2020). The final stage in diagenetic transformation of biosiliceous sediments is the recrystallization of opal-CT into quartz (Zahajská et al., 2020). Whereas radiolarian 'ghosts' are known to occasionally occur in such sediments (e. g., Gombos, 1984), diatoms are usually obliterated. In addition to burial diagenesis, siliceous microfossils can be impacted by reverse weathering (the formation of authigenic clay by reverse weathering involves the alteration of biogenic silica) and pyritization (i.e., precipitation of pyrite minerals in microfossil skeletons) (Michalopoulos and Aller, 1995; Pellegrino et al., 2023).

This compilation of deep-sea diatom-bearing sediments therefore covers all types of marine sediments containing as little as traces of diatoms, up to lithologies composed mainly of diatoms, such as diatomites. Cherts have also been included if they are documented as originating entirely or partially from diatoms.

### 2.2 Site selection

The compilation is based on DSDP, ODP and IODP reports. These international programmes are well documented, with consistent reporting standards, and cover the world's oceans, with 1624 sites drilled to date. No other type of drilling programme has been added to the compilation, due to the lack of homogeneity of the published information.

This literature survey comprises 189 sites in the Arctic, Atlantic, Pacific, Indian and Southern oceans (see Figus et al., 2024b and Fig. 7). Due to the difficulty of determining the palaeolatitude of the Southern Ocean, the high-latitude sites in the Southern Hemisphere are divided by basins and included in the data for the Atlantic, Pacific and Indian Ocean.

Only sites considered to be deep-sea locations are selected, in order to investigate differences in diatom deposition mechanisms between shallow shelf environments (Figus et al., 2024a) and deeper oceanic settings. The boundary between shallow and deep marine environments is set here at a water depth of 200 metres. Hence, continental slopes are divided between shallow (referred to as 'open ocean setting diatomites' in Figus et al., 2024a) and deep marine diatom-bearing sediments (see summary sketch in Fig. 1). Consequently, DSDP, ODP and IODP sites where diatoms were deposited in shallow waters, such as sites 339, U1360 or U1567, are not included in this study.

The second parameter limiting the compilation is the age of deposition. Only Palaeogene deposits (66 to 23 Ma) are included in this compilation, binned into intervals with a 1 million-year (Myr) resolution. The ages are based on biostratigraphic studies carried out by the drilling expeditions, using assemblages of nannofossils, foraminifera, diatoms, radiolarians or dinocysts present in the sediments. All the ages in this study are given on the Gradstein et al. (2012) timescale.

Despite an overall good age determination, a few sites (notably of the older DSDP programme) were rejected due to insufficient age control (i.e., Sites 118, 267, 268, 270 and 674) reported in the literature.

The sites palaeolatitudes in each time bin (see Fig. 2e) were computed with GPlates (Müller et al., 2018) using the Torsvik et al. (2019) rotation model (Figs 3–5). The sites palaeobathymetry in each time bin (see Fig. 2c) was then calculated by extracting the depth from Straume et al. (2024) palaeotopographic maps at the palaeocoordinates computed above. Given those maps resolution and the uncertainty of their underlying model, the reported palaeodepths are thus a somewhat broad estimate of the bathymetry near the sites.

## 2.3 Sampling normalization

In order to provide a baseline for the geographic distribution of the diatom-bearing sediments, we have also compiled a list of all the DSDP, ODP and IODP deep-sea sites containing Palaeogene sediments, regardless of their lithology. To do so, we use the site age profiles, as compiled in the National Geophysical Data Center for DSDP Legs 1 to 96 and ODP Legs 101 to 129 (National Geophysical Data Center, 2000, 2001), in the Janus database for ODP Legs 178 to 201 (Mithal and Becker, 2006)), and as reported in the Preliminary Results of ODP Legs 130 to 177 and IODP Expeditions 301 to 402. The ages provided by those various sources are limited in resolution, as they are provided in the form of Epoch/Stage names rather than a numerical age. The resulting dataset can be found in Supplementary material.

This baseline is also used to normalize the number of diatom-bearing sites seen in the deep-sea: a classical rarefaction (Sanders, 1968) is applied to the number of sites drilled during each time bin (with a quota of 90 sites), and to the number of sites drilled per basin (with a quota of 25 sites) during each time bin, over 10 000 trials. Fig. 6 shows the median number of diatom-bearing sites over each trial, as well as its interquartile range.

## 2.4 Comparisons with other proxies

To provide context to the diatom dataset and to allow us to understand any discrepancy between our collection of diatom-related cherts and the timeseries of chert occurrences reported in Muttoni and Kent (2007; Fig. 2), we updated the dataset of radiolarian abundances reported in Renaudie (2016), based on smear slide descriptions from DSDP Legs 1 to 96, ODP Legs 101 to 129, 178 to 201 and IODP Expeditions 206 to 308, by adding sites for which new age models were available in the Neptune Database (Renaudie et al., 2023).

Finally, in order to determine the possible factors influencing diatom distribution, we compared our results with several geochemical proxies (Fig. 8). Chemical ratios like $^{87}Sr/^{86}Sr$ and $\delta^7Li$ (strontium and lithium isotopes, respectively) track the degradation of continental rocks by interaction with $CO_2$ and water, combined to form carbonic acid ($H_2CO_3$). This reaction consumes atmospheric $CO_2$ and releases the anions and cations present in the weathered rock, such as dissolved silica ($H_4SiO_4$ Froelich and Misra, 2014) or phosphorus (P Föllmi, 1995). Records of $p_{CO_2}$ (Fig. 2a) and silicate weathering are therefore closely related, as are phosphorus and deep-sea diatom-bearing sediments (Fig. 8d, e). Nitrogen, on the other hand, comes from more sources than P, such as continental weathering, nitrate upwelling, or the atmosphere (Tyrrell, 1999). Bioavailable nitrogen can be supplied to the oceans by organisms like cyanobacteria, capable of capturing dinitrogen ($N_2$) from the atmosphere or their environment, and fixing it into ammonium ($NH_4+$) (Kast et al., 2019). Nitrification by microbes oxidizes ammonium to nitrite ($NO_2-$) by ammonia-oxidizers and nitrite to nitrate ($NO_3-$) by nitrite-oxidizers (Bernhard, 2010). Phytoplankton and bacteria can then use these bioavailable forms of nitrogen. The degradation or grazing of living phytoplankton such as diatoms, as well as the reduction of nitrate to dinitrogen by oxidation of organic matter (denitrification), enable the recycling of N forms (Tyrrell, 1999; Bernhard, 2010). Denitrification increases the $^{15}N/^{14}N$ ratio ($\delta^{15}N$) of the water column, due to the preferential uptake of $^{14}N$, creating a spatial gradient in the water column (Kast et al., 2019). A stratified ocean is therefore expected to have a high $\delta^{15}N$ According to Litchman (2007), higher nitrogen concentrations increase diatom abundance, but water column stability (e.g., ocean stratification) has the opposite effect, meaning that the $\delta^{15}N$ record and diatom-bearing sites number should be negatively correlated.

## 3 Results and interpretation

### 3.1 Impact of biases on the observed pattern

In order to be able to draw comparisons with environmental parameters, it is first necessary to consider potential biases in the deep-sea diatom-bearing sediment data.

#### 3.1.1 Sediments preservation

As discussed in section 2.1, diagenetic factors are very likely to introduce an inherent bias into our compilation. Consequently, even though we have considered all the concentrations of diatoms in the sediments in our compilation, including traces, the possibility of a bias arises. If diatoms were undetectable in the cherts drilled and analyzed by the shipboard scientists during

the drilling expeditions, and thus not included in their reports, these cherts would not have been included in our compilation (see section 2 Material and methods for further explanation on site selection). In order to verify this option, a distinction is made between cherts and other sediments in each ocean (except the Arctic, for which no chert occurrence records were found). These results are compared with those of Muttoni and Kent (2007) in Fig. 7. From the middle Eocene onwards, our results are similar, but those from the Palaeocene and early Eocene differ. Considering that Muttoni and Kent (2007) include all types of cherts, while our compilation is based on diatoms only, two hypotheses can be put forward: 1) some sites are absent from our compilation due to the impossibility of distinguishing diatoms and radiolarians in the cherts, or 2) in the Palaeocene-early Eocene, most cherts were derived from radiolarian skeletons. The answer remains indeterminable, owing to the dissolution of siliceous microfossils, and the presence only of preserved radiolarians in a chert does not exclude the likelihood that originally, diatoms may also have been present. Diatoms are dissolved more easily than radiolarians, as they are less heavily silicified, due to their size and surface/volume ratio (Hein et al., 1990). This difference affects the time it takes for the particles to sink (Stoke's law). Particles that sink slowly and have a higher surface ratio, have a greater reaction potential.

### 3.1.2 Sampling bias

The various deep-sea drilling campaigns did not recover sediments of all ages at an equal rate: in particular, the early Palaeogene (early Palaeocene to early Eocene) was significantly undersampled compared to the subsequent stages (see Fig. 7). Correcting for this bias using rarefaction, we show that while most of the time series pattern in each basin, as well as globally, remain mostly unchanged and consistent from the end of the Early Eocene onwards, the previous time interval does show significant changes, rather than a steep, uniform increase in the number of diatom-bearing sites from the K/Pg boundary to the middle Eocene. What we actually see is that in the middle Palaeocene, diatom-bearing sites are as frequent (proportionally) as they are in the middle to late Eocene, while the late Palaeocene shows an abrupt drop, recovering only at the end of the Ypresian. Whether this early Eocene diatom gap (particularly clear in the Atlantic) is real, or a preservation effect (as it is coeval with a known chertification event in the North Atlantic; e. g., Muttoni and Kent, 2007) remains unclear, though as was shown above, those cherts are potentially not diatom-related, but radiolarian-related.

### 3.2 Impact of nutrients and ocean circulation

### 3.2.1 Early Palaeogene

After the Cretaceous-Palaeogene boundary, the decrease in silicate weathering proxy values in the Palaeocene (Fig. 8) is linked to the prevalence of peneplain landforms (Misra and Froelich, 2012; Froelich and Misra, 2014; Tsekhovsky, 2015). Misra and Froelich (2012) propose two possibilities to interpret the lower $\delta^7$Li values in the late Palaeocene–early Eocene: 1) the input of light Li from rivers to the oceans, in association with the numerous hyperthermal events leading to intensified weathering of cation-depleted peneplain rocks at low latitudes; or 2) the influence of Large Igneous Provinces (LIPs) such as the Deccan Traps or the North Atlantic Igneous Province (NAIP). As basalt is rapidly weathered yielding lighter Li, the position of India with freshly erupted LIPs at the Equator around the Palaeocene-Eocene boundary might have led to the drop in $\delta^7$Li at this

period. A similar process may have occurred with the NAIP during the Middle Eocene Climatic Optimum (MECO; ~40.5 to ~40 Ma), leading to a reduction in the number of diatom-bearing sediments in the North Atlantic (Fig. 8e). Furthermore, this second possibility may explain the two drops in $^{87}Sr/^{86}Sr$ at the EECO and MECO, and would be consistent with the hypothesis of McArthur et al. (2001), supporting the idea of a decrease in $^{87}Sr/^{86}Sr$ linked to the LIPs in the Palaeocene. The phosphorus record (Fig. 8d), which is indicative of continental weathering rates, is also more consistent with this scenario. The $P_{tot}$ curve represents all forms of phosphate, while $P_{bio}$ is a measure of bioavailable phosphorus. Föllmi (1995) explains that the similarity between the two curves indicates that changes in bioavailable phosphorus, originating from chemical weathering, are similar to the total continental weathering. Therefore, although $\delta^7Li$ and $^{87}Sr/^{86}Sr$ are lower in the early Palaeogene, the high accumulation rates of phosphorus are consistent with enhanced continental weathering driven by hyperthermal events (Penman, 2016).

Comparison of the $\delta^7Li$ record with the number of deep-sea diatom-bearing sediments (this study) and the number of shallow marine diatomite occurrences (Figus et al., 2024a) suggests that diatoms were mainly accumulating in the North Atlantic and Eurasian epicontinental basins (in particular the western Siberian marine areas) during the early Palaeogene (Fig. 7), whereas the low and mid-latitudes of the Southern Hemisphere appear to be depleted in this respect (Fig. 9a). Moreover, the number of epicontinental sea diatomite occurrences (Fig. 7) is very similar to $P_{bio}$ (Fig. 8d), demonstrating the influence of nutrients – provided here by continental weathering during hyperthermal events – on diatom deposition. However, despite an increase in the number of diatom-bearing deposits in the Atlantic during the early Palaeogene, none of the deep-sea diatom-bearing records appear to be correlated with phosphorus availability. After correcting for sampling bias (Fig. 6), it appears that early-middle Palaeocene diatom-bearing sediments are probably artificially low, due to the number of drilled sediments dating from this age. Thus, the apparent trend of steady increase from the Cretaceous-Palaeogene boundary is most likely false, but the amount of diatom-bearing sediments show a substantial increase from 48 Ma onwards.

Differences between shallow and deep-sea realms might be explained by ocean circulation. Although nutrients can be easily distributed by rivers in near-shore environments, the influence of transport remains limited (Araujo et al., 2017; Katz et al., 2020). Phytoplankton would therefore bloom mainly near coastal areas and in shallow basins. In the Palaeocene, $\delta^{15}N$ was particularly high in the Pacific before falling abruptly around the PETM onset (Fig. 8c). According to Kast et al. (2019), elevated $\delta^{15}N$ indicates suboxic conditions, presumably due to increased stratification of the oceans during this period. A comparison with the number of diatom-bearing sites in the Pacific suggests an inverse relationship between these two records, which would be consistent with the effect of global ocean ventilation on nutrient distribution and diatom deposition. This hypothesis could also apply to the Indian and Atlantic oceans. While diatom-bearing sediments in the Atlantic are distributed near the margins that provided nutrients in the Palaeocene, the subsidence of the Greenland–Scotland Ridge may have improved ocean ventilation from the Eocene onwards (Vahlenkamp et al., 2018), and allowed the formation of intermediate waters in the North Atlantic around Greenland.

### 3.2.2 Middle Eocene diatomite gap

In Figs 2 and 4, a peak in continental weathering and temperature proxies is observed, as well as a drop in $p\mathrm{CO}_2$, at about the same time as the decrease in the Atlantic diatom-bearing sediment occurrence record, from ~42.5 Ma onwards. These results suggest that global temperatures have increased, and that continental weathering has intensified. In addition, the lowering of $\delta^{15}\mathrm{N}$ (Fig. 8c) is caused by the remineralization of organic nitrogen into nitrate, linked to the preferential uptake of 14N by phytoplankton (Auderset, 2020). Moore (2008) suggests that nutrient and dissolved silica recycling rates may have been intensified by an increase in the metabolic rates of marine organisms, linked to the warm temperatures of the early Eocene (Olivarez Lyle and Lyle, 2006). With the increased weathering and recycling of N in the Lutetian, nutrients were highly available for diatoms during the diatomite gap, but differences in the distribution of P and N influenced the diatom-bearing sediments accumulation. P is distributed predominantly at the margins (e.g. North Atlantic and Northern Indian Ocean), reflecting its weathering-related origin, whereas bioavailable N is supplied by upwelling as in the Central and Equatorial Pacific. Figure 7 compares data obtained in this study with the shallow marine diatomite records of Figus et al. (2024a) and suggests contrasting patterns in diatom deposition between the North Atlantic – as shown in Fig. 4, 9d-e, these deposits are mainly distributed in the North Atlantic – and epicontinental seas during the middle Eocene. In the Pacific and Indian oceans, the number of deep-sea diatom-bearing sediment occurrences increases from the end of the EECO and through the diatomite gap, while the number of diatomite occurrences decreases in epicontinental seas (Fig. 7). In the Pacific, the curve marks a step during the diatomite gap, before abruptly increasing again at the same time as the number of shallow marine diatomite occurrences. Global cooling following the EECO may be responsible for the general increase in the number of diatom-bearing sites in the Pacific, but the sampling bias shown in Fig. 6 suggests that the number of diatom-bearing sites in the Pacific increases with the number of drilled sites in the early to middle Eocene. However, it seems that the peak in diatom-bearing sediment occurrences in the Indian Ocean during the middle Eocene is genuine, like in the Atlantic (Fig. 16). In addition to biosiliceous production, diatom diversity (Fig. 7) is increasing (Lazarus et al., 2014). Sims et al. (2006) and Witkowski (2022) have already reported this explosion in diversity, explaining that numerous genera have appeared, in particular monospecific ones. Figus and Witkowski (2024), for example, describe two monospecific genera from the middle Eocene, discovered in samples from the North Atlantic, which coincide with the peak in diversity and the number of Atlantic diatom-bearing sediments.

Figus et al. (2024a) attribute the diatomite gap between ~46 and ~44 Ma to tectonic reorganization, involving, for example, the closure of the western Siberian Sea strait and the Arctic basin connection, which probably led to the establishment of more terrigenous-dominated sedimentation (Akhmetiev et al., 2010). Inter-basin reorganization of the area may have been responsible for the increase in continental weathering, resulting in lower $p\mathrm{CO}_2$ and enhanced phosphorus inputs. Witkowski et al. (2021) suggest that the Northern Component Water may have improved Atlantic ocean circulation in the middle Eocene, enabling a better distribution of nutrients, and thus intensifying biosiliceous production. It could be hypothesized that biosiliceous production may have also contributed to the decrease in $p\mathrm{CO}_2$, as suggested by Cermeño (2016): the increase in nutrient availability as a result of enhanced silicate weathering would have led to the ecological rise of diatoms, which participate through

photosynthesis in the release of organic carbon that will be buried in sediments. However, the resolution of our dataset is too coarse to provide an answer.

### 3.2.3 Latitudinal and inter-basin shifts in the distribution of diatom-bearing sediments

A close examination of the occurrences of shallow marine diatomite deposited in open ocean conditions (i.e., deposited in the first upper 200 metres of water depth on continental slopes, see Figus et al., 2024a) suggests a similar pattern within the Pacific and Indian Ocean curves, although the number of shallow marine diatomite occurrences appears to increase as the number of deep-sea diatom-bearing sediment occurrences decreases, and vice versa, from the Priabonian onwards (Fig. 7). Interestingly, almost all the diatomite occurrences in open ocean settings in Figus et al. (2024a) are distributed around the Pacific (i. e. Kamchatka and the west coasts of North and South America), starting with a high number of diatomite occurrences along the west coast of North America in the Bartonian, followed by increased accumulation of diatomite on the western continental slope of South America in the Priabonian, before the onset of the deposition of shallow marine diatomite in the Kamchatka region in the Oligocene.

Despite the general picture of a sluggish Eocene ocean (Haq, 1981; Keller, 1983; Barron, 1987; Miller, 1992; Huber and Sloan, 2001), Moore (2008) suggests that there is no evidence of weaker Eocene thermohaline circulation, and that only wind-induced ocean currents have been slowed. Consequently, the increase in the number of deep-sea diatom-bearing sediment occurrences, and later shallow marine diatomites, in the Pacific and Indian oceans, may be the reflection of improved ocean ventilation. Moore et al. (2008) present results on the recovery of Eocene radiolarian zones in several basins, which display a similar pattern to our results on the number of diatom-bearing sediments (Fig. 7), showing an apparent shift from the Atlantic to the Pacific and Indian basins in the late Eocene. This change in the distribution of siliceous microfossils around 43 Ma is interpreted as the result of a shift in the formation of deep ocean waters, from the North Pacific to the Southern Ocean (Thomas, 2004; Moore et al., 2008). Furthermore, Fig. 4 and 9 show that around the same time, the distribution of the diatom-bearing (Fig. 9a) and radiolarian (Fig. 9b) deposits changed from mid-northern to tropical latitudes.

It has long been a widely repeated statement that as the ocean cooled through the Palaeogene, the Southern Ocean progressively became the key diatom accumulation locus. This perspective seems to have been formulated in the early days of the DSDP era (e. g., Fenner, 1984, 1995), which shaped our overall understanding of diatom-bearing deposit distribution in deep time. Bryłka et al. (2024) challenged such views, and showed that there was very little quantitative support for whatever overall statements on how diatoms may have reacted to the rapid cooling at the Eocene-Oligocene Transition (EOT; ~33.9 Ma). While the present study also does not provide a quantitative perspective on diatom accumulation, our data may be interpreted as a measure of geographic distribution of diatom preservation in sediments, with an underlying caveat that seafloor preservation reflects the surface water production rates.

Figure 2d-e shows cumulative plots of the distribution of diatom-bearing sites through time by latitude and by ocean basin. These indicate that through our entire study interval, the tropics and mid-northern latitudes have consistently witnessed the most widespread deposition of diatom-bearing sediments (Fig. 3, 4, 5 and 9). The contribution of northern high latitude and mid-southern latitude zones is negligible, but the distribution of drilling sites in the Atlantic Ocean shows a substantial bias

against the Southern Hemisphere (Witkowski et al., 2020), and so this perspective may be inaccurate. The southern high-latitude belt shows higher percentages in the Palaeocene, when diatom-bearing sediments are generally sparse (thus making a biased view more likely), and in the late Eocene-Oligocene interval (Fig. 2e, 4, 5). In the latter case, however, the increase takes place not at the Eocene-Oligocene boundary, but at the onset of the late Eocene (~37.5 Ma).

When viewed by ocean basin, we note that through the Palaeocene and early Eocene (except for a brief early Palaeocene interval of increased Pacific contribution), diatom deposition is largely focused in the Atlantic Ocean, consistent with the observations summarized above and in previous studies (e. g., Witkowski et al., 2020) (Fig. 2d, 3, 4). From around the Palaeocene-Eocene boundary, and especially so from the termination of the EECO at ~49 Ma, the Indian Ocean and the Pacific show a progressively higher proportion of diatom-bearing sites relative to the Atlantic. These proportions largely stabilize at ~41 Ma and remain so, with subordinate fluctuations, until the end of the Palaeogene. Notably, there is no drastic inter-basinal shift at the Eocene-Oligocene Transition. Thus, again, if our data are to be treated as a measure of geographic extent of biosiliceous production, the pattern of diatom-bearing site distribution largely opposes the traditional view of biogenic silica accumulation being progressively more focused in the Southern Ocean in the latter half of the Palaeogene.

## 3.3 Impact of glacial events and Antarctic Ice Sheet setting

A peak in the number of diatom-bearing sites occurs at ~37.5 Ma in the Pacific and Indian oceans (Fig. 7). Fontorbe et al. (2017) describe an enrichment of Pacific equatorial waters in dissolved silica from 37 Ma, in relation to the widening of the Drake Passage and the deepening of the Tasman gateway, affecting ocean circulation. However, Pascher et al. (2015) report the response of radiolarians to a glacial event in the Pacific: the Priabonian Oxygen Isotope Maximum (PrOM; ~37.3 Ma). This event is associated with a positive oxygen isotope excursion and a continental-scale glaciation of Antarctica (Van Breedam et al., 2022). This early ice sheet expansion might have enhanced continental weathering, contributing to phosphorus (Fig. 8d) and dissolved silica enrichments (Hawkings et al., 2017) that led to increases in the abundance, preservation and diversity of radiolarians (Pascher et al., 2015) and diatoms (Fig. 7).

Similarly, the number of diatom-bearing sediment occurrences in the Pacific and Indian oceans, as well as diatom diversity (Fig. 7), appear to respond to the Oligocene glacial events Oi-1 (~33.6 Ma; Pälike et al., 2006) and Oi-1a (~32.8 Ma; Galeotti et al., 2016), which mark the onset of a permanent Antarctic Ice Sheet. The perturbations of the climate and Antarctic Ice Sheet are closely related, with retreats of the ice sheet during warmer periods, and its expansion during cooler periods (Zachos et al., 2001) such as the global cooling set at the Eocene-Oligocene Transition. The fact that the number of deep-sea diatom-bearing occurrences in the Pacific and Indian oceans does not peak at exactly the same time (first the Indian Ocean with Oi-1, then the Pacific with Oi-1a) might be a problem with the resolution of our data, or may be related to a basin-to-basin fractionation of deep-sea biogenic sediments. Witkowski et al. (2021) reported several shifts in biogenic silica fluxes between the Pacific and Atlantic, around 42, 37 and 35 Ma. These shifts are all illustrated in Fig. 7, and the decoupling between the Pacific and the Atlantic may be explained by the fractionation of biogenic sediments by deep ocean circulation (Berger, 1970; Witkowski et al., 2021), as explained in section 3.2.3. Another peak in the number of deep-sea diatom-bearing sediment occurrences is observed in each basin after the Oi-2 (~30.35 Ma; Wade and Pälike, 2004), another Oligocene glacial event. Due to the differences of

resolution between the dating of these climatic events and our results, the cause of the increased number of diatom-bearing deposits remains uncertain, but the presence of a permanent Antarctic Ice Sheet impacting continental weathering and nutrient distribution during this period appears to be a plausible explanation.

## 3.4 Arctic Ocean

Only four of the sites drilled by ODP and IODP in the Arctic Ocean contain Palaeogene diatom-bearing sediments. As a result, the data are not representative enough of the Arctic basin to be interpreted in this study. Nevertheless, these sites have been retained in the compilation to ensure the global representation of deep-sea diatom-bearing sediment distribution. Figure 7 illustrates that diatom-bearing sediments started accumulating at our Arctic sites during the EECO, which could be interpreted as an effect of global cooling (Fig. 2b). Additionally, the Arctic curve indicates a slight increase in the number of
diatom-bearing sediments during the diatomite gap, simultaneously with the step marked by the Pacific curve, and during the Oi-2 glacial event. Ocean circulation, and indirectly climate, might therefore explain these peaks, as in the other oceans. This remains, however, a hypothesis, and it would be necessary to study a larger number of diatom-bearing sites in the Arctic to formulate an interpretation.

## 4 Conclusion

Despite the high number of drilling expeditions which covered the world's oceans, compiling deep-sea diatom-bearing data is challenging, due to the unequal temporal and spatial coverage of the sediments. In addition, the diagenetic processes occurring with increasing burial depths impact the preservation of siliceous microfossils, making their identification more difficult in the oldest sediments. However, we present the first comprehensive study based on all Palaeogene deep-sea diatom-bearing sediment occurrences at DSDP, ODP and IODP sites. Compiling these data allowed us to determine that nutrient availability and ocean
circulation primarily control the number and distribution of deep-sea diatom-bearing sediments. The results of Figus et al. (2024a) suggest that ocean circulation and tectonic reorganizations are the main drivers of shallow marine diatomite deposition, but tectonics seem to have only an indirect impact on the number of deep-sea diatom-bearing occurrences (particularly in the Atlantic) during the diatomite gap (~44 to ~46 Ma), through intensified silicate weathering leading to increased nutrient availability. Climate also appears to have only an indirect correlation with the deposition of diatom-bearing sediments in both
shallow and deep marine environments, impacting deep-sea diatom-bearing sediments in the Pacific and Indian Ocean as a result of the extension of the Antarctic Ice Sheet during the Oligocene global cooling. Our synthesis also shows that through most of the Palaeogene, low to mid-latitude areas, especially in the Atlantic Ocean, were the locus of widespread biogenic silica deposition and burial. This challenges the traditional view that assumes the Southern Ocean became the key biogenic silica sink in the late Eocene, and has remained so ever since.

*Data availability.* The compilations of deep-sea diatom-bearing sediments, radiolarian occurrences and age profiles can be found on Zenodo (Figus et al., 2024b). Palaeogeographic maps showing all drilled sites in each 1 Myr time bin can be found in the Supplementary Materials.

*Author contributions.* JW designed the study. CF and JW prepared the compilation. JR processed the data. All co-authors participated in the interpretation of the data. CF prepared the manuscript, with contributions from all co-authors.

*Competing interests.* The authors declare that they have no conflict of interest.

*Acknowledgements.* We would like to thank Eunah Han and one anonymous reviewer for their comments on this paper.

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

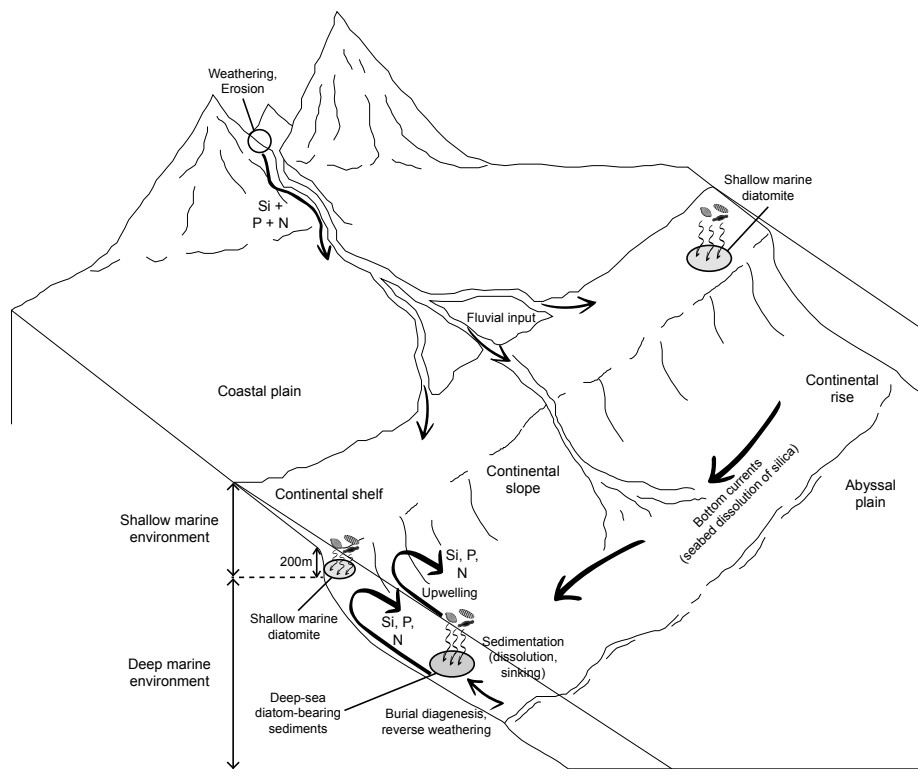

**Figure 1.** Summary of the impact of ocean circulation and nutrient availability on the deposition of diatom-bearing sediments.

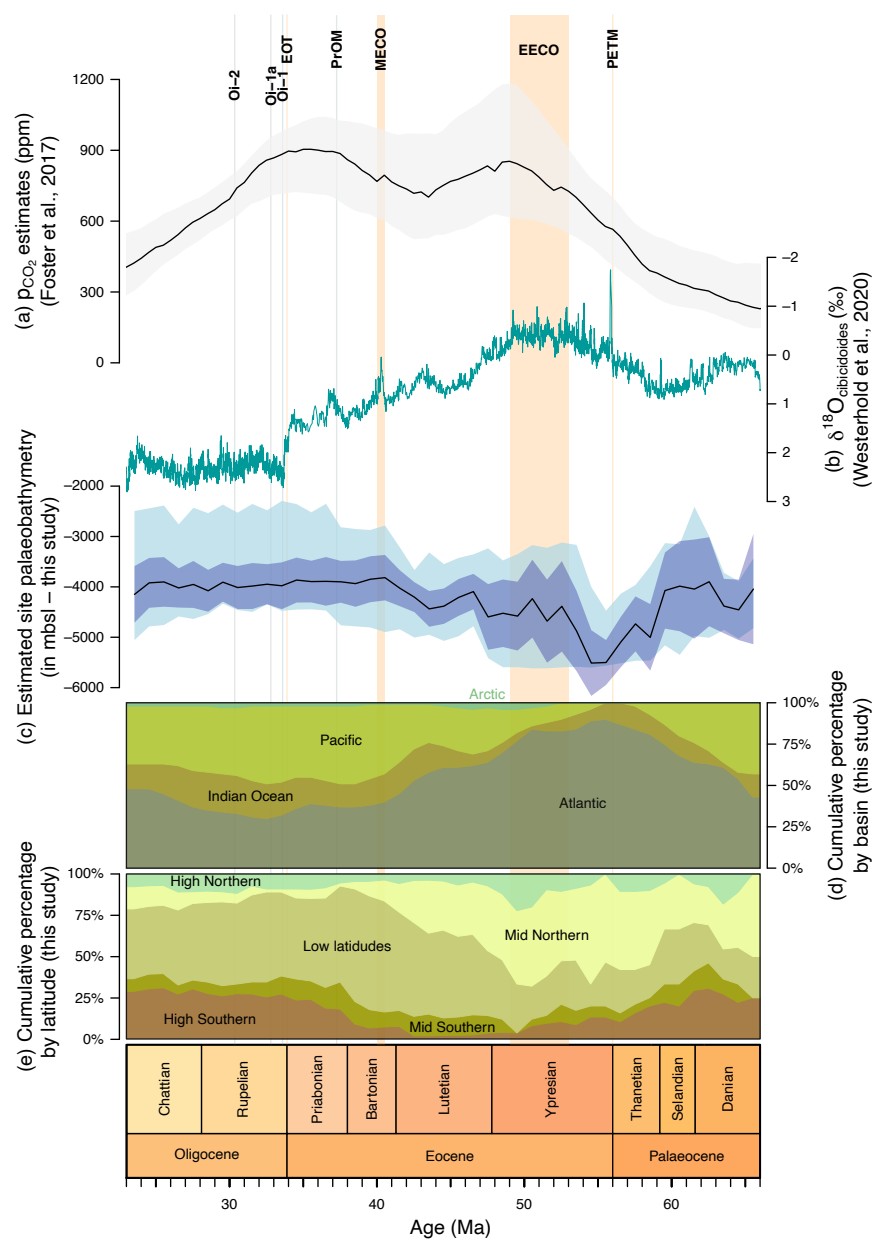

**Figure 2.** Comparison between our data and various proxies. (a) $p_{CO_2}$ estimates from Foster et al. (2017), with mean in black and 95% confidence interval in grey; (b) $\delta^{18}O_{Cibicidoides}$ from Westerhold et al. (2020); (c) median estimated site palaeobathymetry (msbl) with 95% confidence interval on the median (McGill et al., 1978; in dark blue) and interquartile range (Tukey, 1977; in lighter blue); (d) cumulative percentage of diatom-bearing sediment occurrences by basin (this study); (e) cumulative percentage of diatom-bearing sediment occurrences by palaeolatitude (this study), with limits defined from -90° to -45° and from 90° to 45° for the high latitudes, from -45° to -30° and 45° to 30° for the mid latitudes, and from -30° to 30° for the low latitudes.

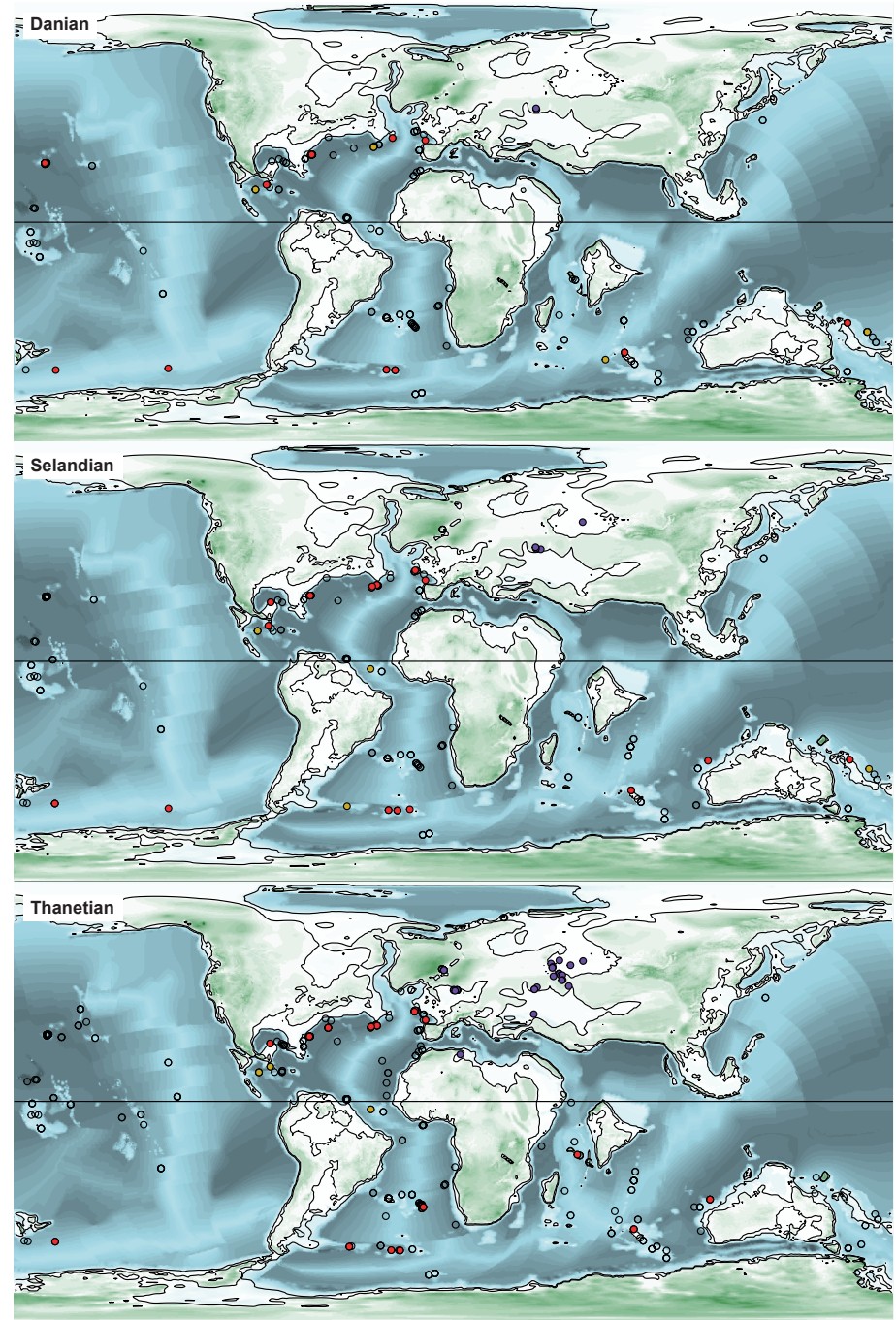

**Figure 3.** Palaeogeographic and palaeobathymetric maps representing all sites in the Danian, Selandian and Thanetian (Palaeocene). The unfilled circles show sites that did not contain deep-sea diatom-bearing sediments at these times. The magenta and grey circles represent shallow-marine diatomites, deposited respectively in epicontinental seas and open ocean settings. The red circles are deep-sea diatom-bearing sediments and the yellow circles are sites containing cherts with diatoms. The palaeobathymetry is from Straume et al. (2024).

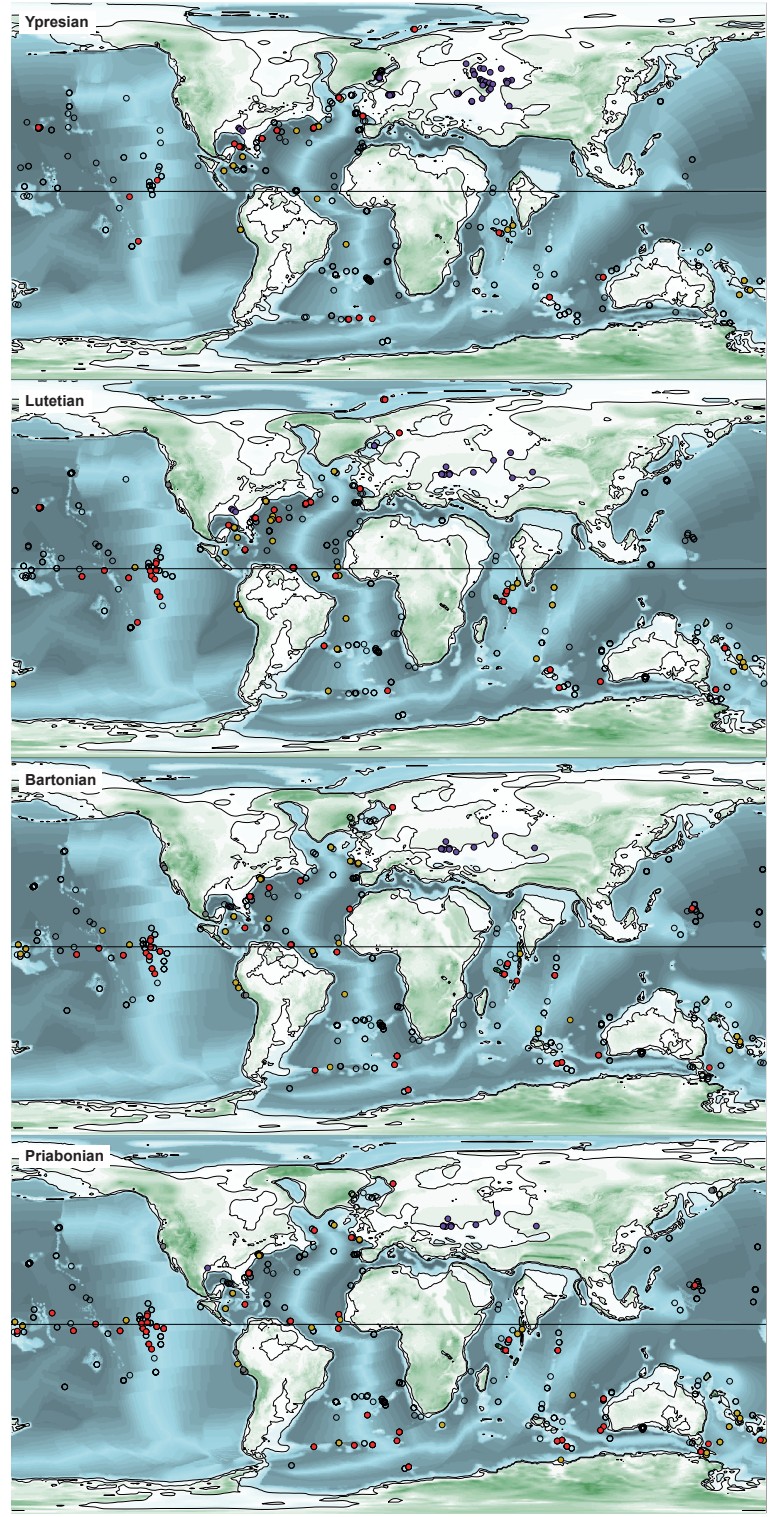

**Figure 4.** Palaeogeographic and palaeobathymetric maps representing all sites in the Ypresian, Lutetian, Bartonian and Priabonian (Eocene). See Fig. 3 for the legends.

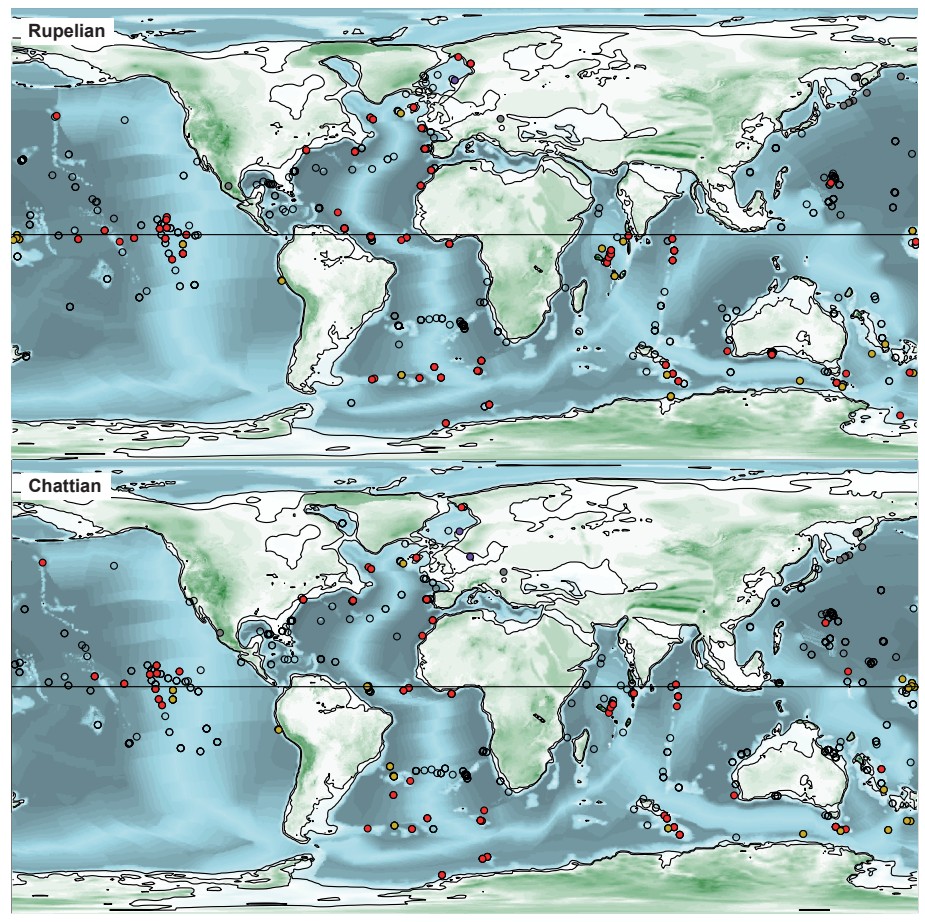

**Figure 5.** Palaeogeographic and palaeobathymetric maps representing all sites in the Rupelian and Chattian (Oligocene). See Fig. 3 for the legends.

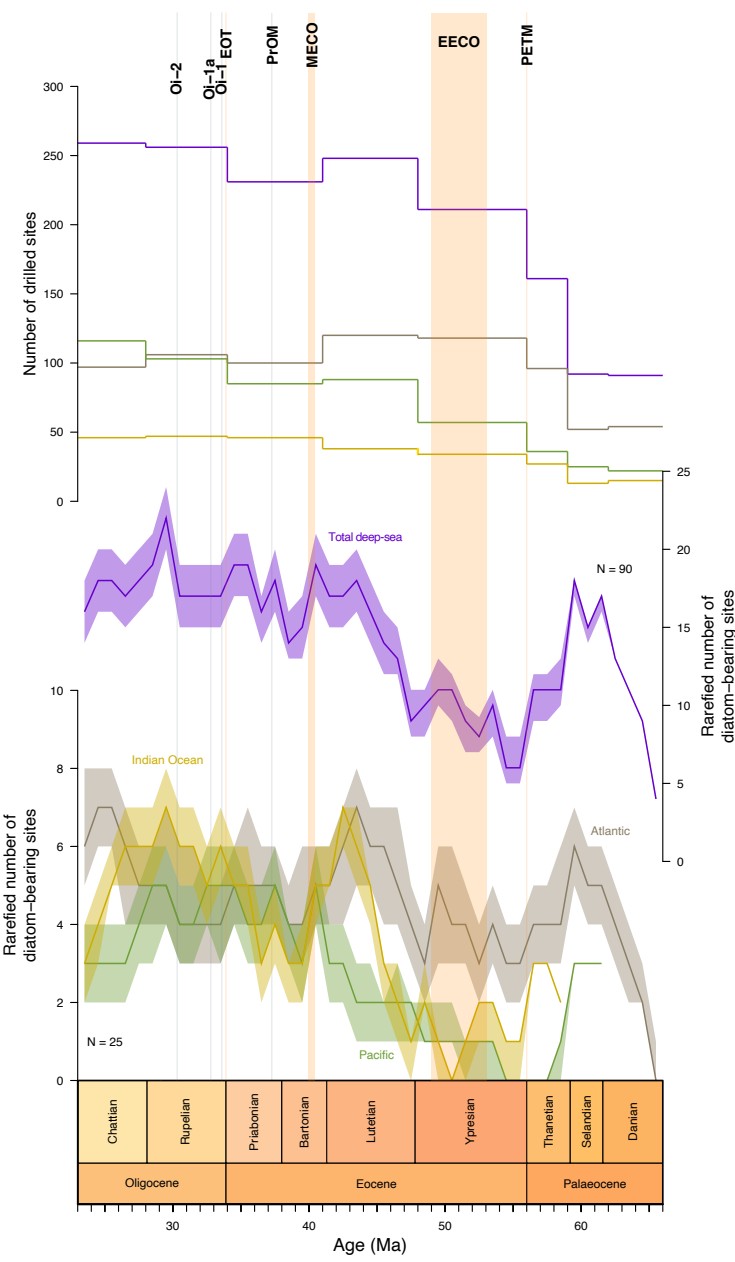

**Figure 6.** Correction of the sampling bias related to the number of drilled sites.

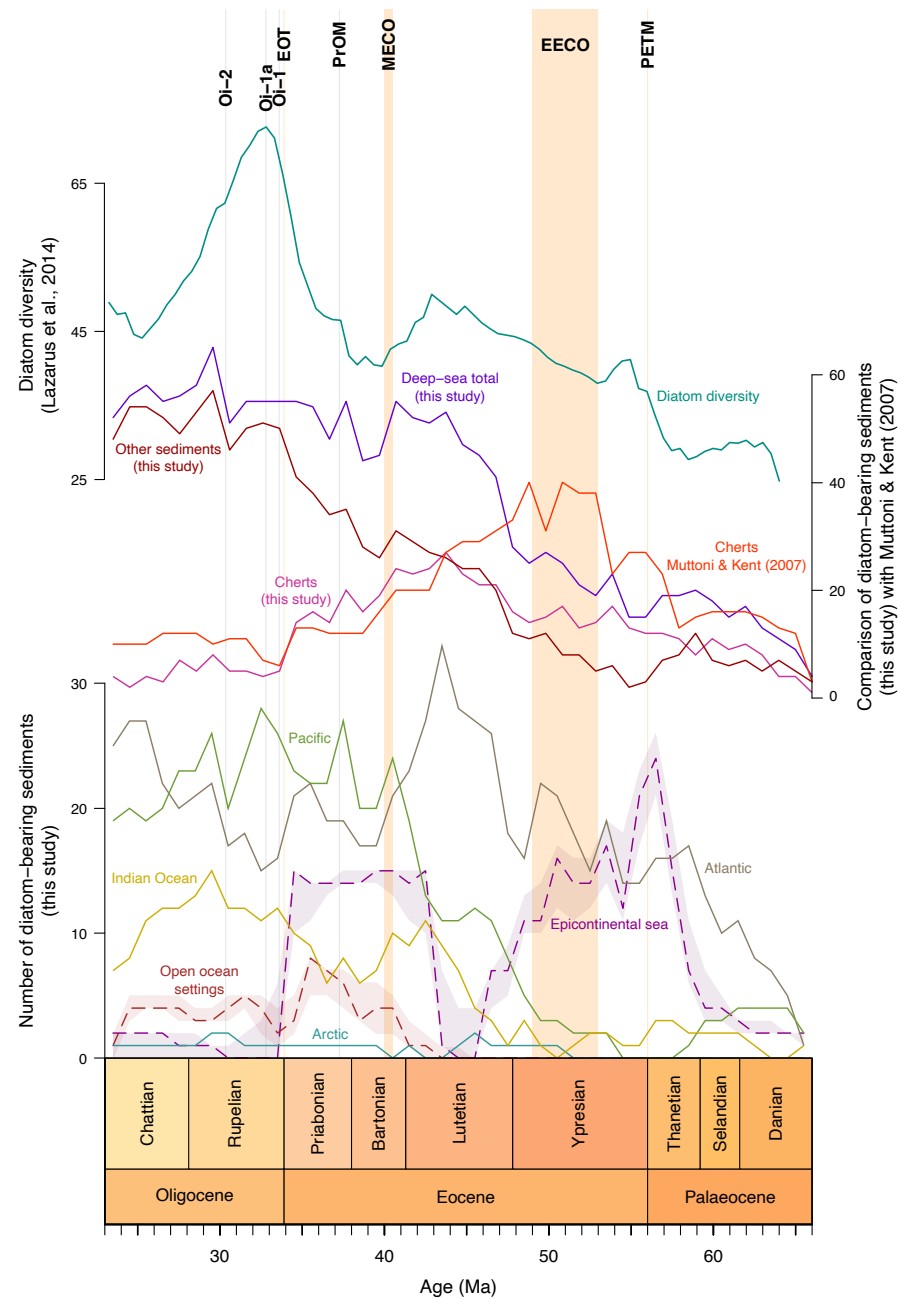

**Figure 7.** Comparison between our data and shallow marine diatomites (Figus et al., 2024a), cherts (Muttoni and Kent, 2007) and diatom diversity (Lazarus et al., 2014) distributions through the Palaeogene.

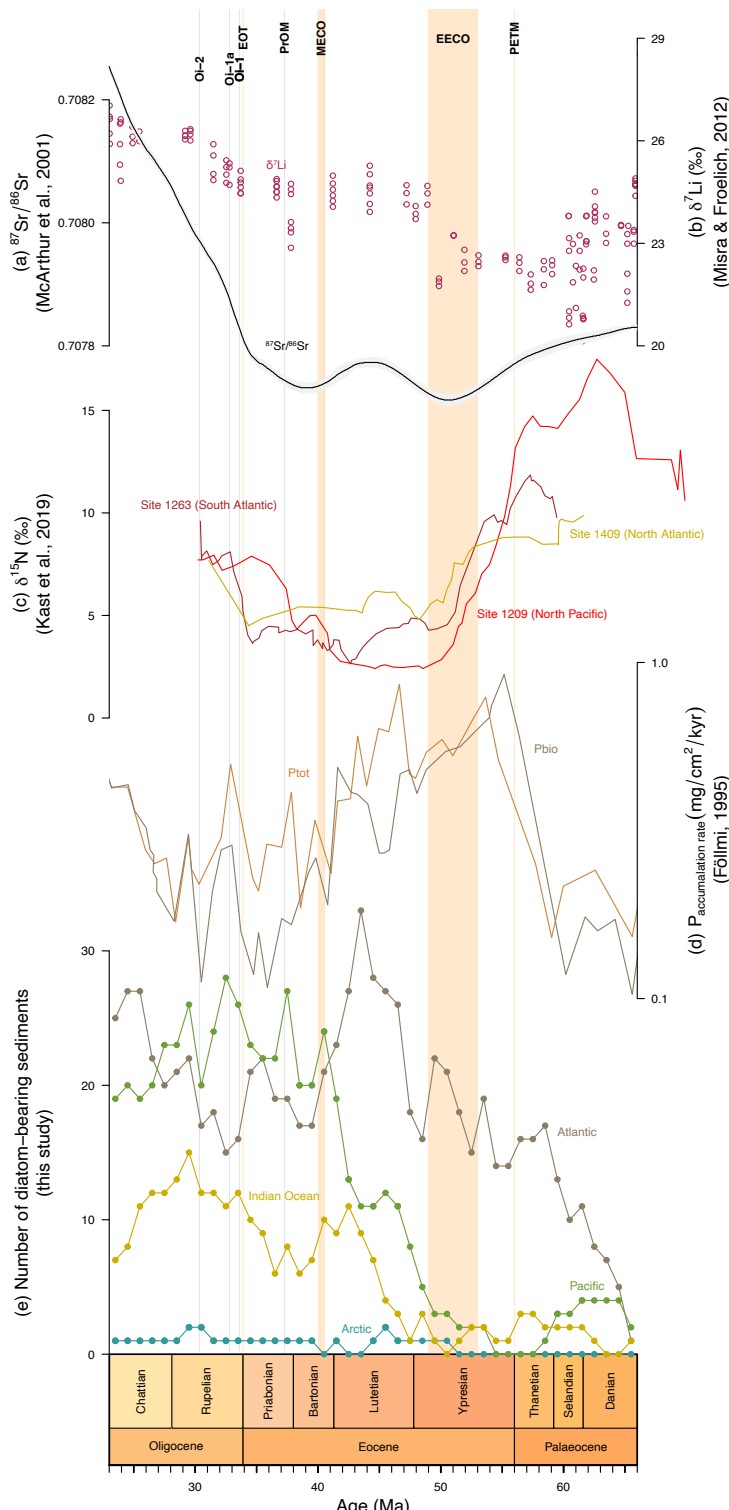

**Figure 8.** Comparision between the number of diatom-bearing sediments and various proxies. (a) $^{87}Sr/^{86}Sr$ from McArthur et al. (2001); (b) $\delta^7Li$ (‰) from Misra and Froelich (2012), (c) $\delta^{15}N$ (‰) from Kast et al. (2019); (d) accumulation rate of phosphorus (P) in $mg.cm^{-2}.kyr^{-1}$ from Föllmi (1995), with $P_{tot}$ representing all types of phosphate sediments and $P_{bio}$ the sediments containing bioavailable phosphorus only; (e) number of diatom-bearing sediments by basin (this study).

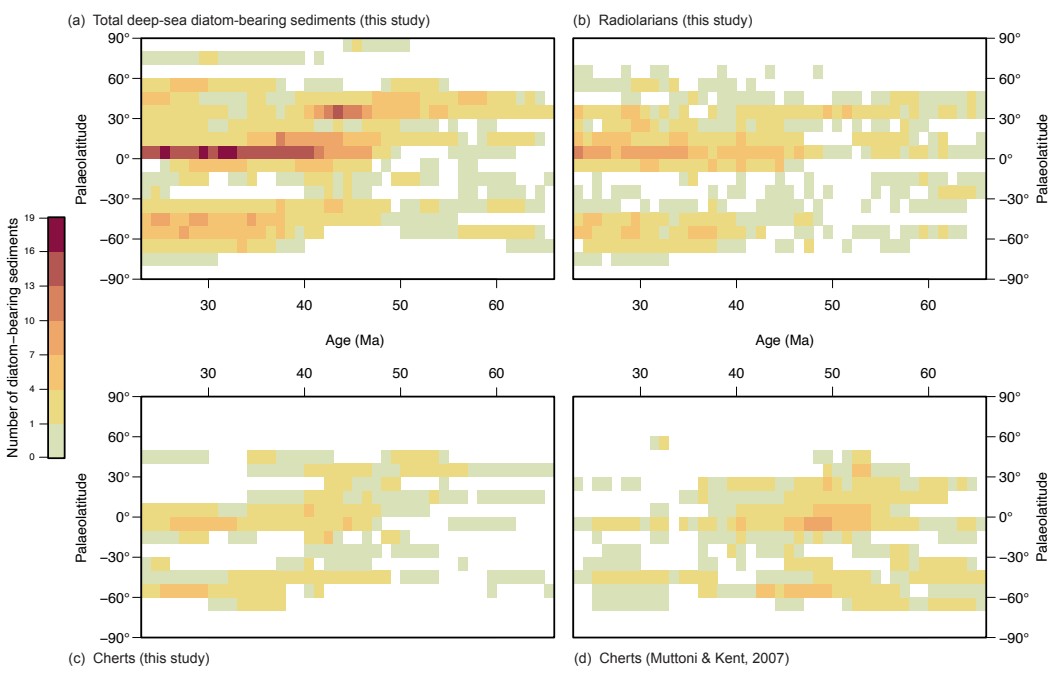

**Figure 9.** Comparison between the distribution of (a) total deep-sea diatom-bearing sediments (this study); (b) radiolarians (this study); (c) cherts from this study and (d) cherts from Muttoni and Kent (2007); by palaeolatitudes.