# Peer review of "Controls on Palaeogene deep-sea diatom-bearing sediment deposition and comparison with shallow marine environments"

_EGUsphere, 2024_

## Referee Comment (RC1)

**Compilations of Palaeogene deep-sea diatom-bearing sediments and associated data**

C. Figus et al., Biogeosciences

**General Comments**

This paper offers a paleo-oceanographic insight into the importance of nutrient availability and ocean circulations versus climate on the abundance of diatoms in water columns. This study's wide spatial and temporal scope broadens its implication, encompassing four of the five oceans from the late Mesozoic to the Cenozoic periods. Despite this strength, I am not convinced about the novelty of this paper. Rather, I see it in terms of a summary review rather than an original research article. This is because most of the key data in this paper is pulled from previous literature. Most newly displayed datasets in this paper are analytically rudimentary based on enumeration (mostly the number or frequency of diatom-bearing sediments). This means that insufficient evidence has been presented to support new findings as it stands. Section 3.2. in the Results and Interpretation section is the core part of this paper, however, it has been augmented with a summary of the previous findings. The author's tone fails to convincingly convey the proposed new ideas (e.g., words such as "probably", "might", or "may" are used when stating new insights). The writing style of this paper is better suited to a technical report rather than a journal paper, focusing on the description of data or findings instead of the critical assessment or the logical connection between them. Therefore, I recommend the authors position this manuscript as being a foundation for another original study, instead of being an independent journal publication. Alternatively, this paper can be condensed into a one-page "Perspective" article, assuming that the main premise behind this study is critical and timely for the current paleo-sciencific readership.

**Specific Comments**

1. L9-10: Please clarify if the diatomite gap was observed in shallow marine or deep-sea sediments. The diatomite gap typically refers to the deposition in shallow marine sediments. This is essential to follow your logical reasoning that the tectonic reorganizations led to the diatomite gap, despite the deep-sea diatom deposition.

2. L14-28: This part lacks any plausible explanation about how tectonics could have affected the deep-sea diatom deposition, although it does describe a potential impact of the other two factors, climate and ocean circulations.

3. I cannot see any sign that the sampling bias correction method in this study has been significantly advanced, nor that it is novel when compared to the existing statistical methodology.

    a. Section 3.1 Impact of Biases on the Observed Pattern in the Results and Interpretation section may not merit its own independent subsection.

    b. Figure 3 could be included as supplementary information, not as a main figure.

4. This paper fails to (i) succinctly introduce a scientific principle or known facts and (ii) connect it to support the original dataset. The latter part (ii) is much more important than the former (i) for a research article. Also, I note that some sentences are logically incomplete. For example, in L145-149:

    a. In general, $N_2$-fixing microbes and diatoms occupy distinctive spatial or temporal niches, oligotrophic and warm waters vs. nutrient-rich cold waters. The statement in L145-149 reads as if $N_2$ fixers and diatoms dwell close to each other and exchange nutrients. This issue can be partially solved by elaborating on "then nitrite (NO2-) and nitrate (NO3-) by nitrification" in a new separate sentence. Were the study areas unaffected by other N sources such as terrestrial input and nitrate upwelling? You may want to add more references about known and potential N sources of the time span and specific sites, in addition to Kast et al. (2019).

    b. Please clarify whether you are referring to benthic diatoms or planktonic diatoms. Their morphology, species, and the $\delta^{15}N$ values all differ.

    c. The sentence in L147-149 can be made clearer by stating what phytoplankton vs. bacterial decomposers do in N cycling, rather than distinguishing phytoplankton vs. diatoms and other organisms. "Used" and "recycled" does not sound mutually exclusive, and "other organisms" is ambiguous.

    d. Please add a sentence about what proxies you analyzed or cited for N cycling, as you did for the Sr and Li isotopes. You can inform (i) how the analysis is related to diatom accumulation (e.g., lower/higher $\delta^{15}N$ of sediments suggests more/less diatom deposition) and (ii) if you expected N dynamics to be related to $pCO_2$, as Sr, Li, silicate, and P all are.

5. The authors state the aim of this study is to identify the factors determining the vertical flux of diatom deposition. In my opinion, the actual goal should be to advance learning in paleoenvironmental conditions, such as nutrient distribution, ocean circulations, plate tectonics, or climate, based on the diatomite data. Thus, I would expect a revised manuscript to discuss the broader implications of the paleoenvironment.

**Technical Corrections**

1. Please be more direct and specific at the individual sentence level. For example:

   a. The Abstract section overuses vague expressions such as "response to", "is mainly controlled by", "an indirect correlation", "linked to", and "a comparison of". Clearer, on-point wording, such as increase/enhance/elevate/intensify or decrease/suppress/prevent/weaken, can be substituted for some of these ambiguous expressions. Adding an adverb or an adjective would help inform the scale of change, such as largely, slightly, or x-fold.

   b. L33-34: The summary of Figus et al. (2024a) should be more direct. How did climatic and tectonic factors and ocean circulations change diatomite accumulation? Please specify the factors and the changing direction (e.g., increase or decrease).

   c. The writing from Section 3.2.1 onwards is clearer than the earlier part.

2. L26: "more" seems unnecessary.

3. L25-47: Please break this paragraph up into two or three paragraphs.

4. L28-31: The second sentence is redundant in terms of the point made in the preceding one. You can combine these two sentences together and add Brylka et al. (2024) to support the limitation of the datasets.

5. L28-30: Did the "several studies" investigate different parts of the oceans or the same location? Please specify which it was in this sentence.

6. L34-35: "the presence of a gap" can be just "a gap".

7. L140: The opening sentence can be more generic, summarizing your approach rather than describing a figure. You can amend this sentence accordingly: "We analyzed x (e.g., four) geochemical proxies to reconstruct the distribution of diatoms (Fig. 4)." You cannot use this sentence if those proxies were analyzed for other papers, rather than for this one.

8. Please make sure that the y-axis of Figure 2(b) covers the entire range of the line graph. The current axis does not encompass a range of higher values greater than 2 ‰.

9. Please add the unit of the y-axes in the following figures: The first two y-axes in Fig. 1; Fig. 2b; Fig. 4a.

10. You can add a map of sampling sites with sampling frequency if you want to include some graphic information about the geographic coverage of the sampling sites.

---

## Author Response (AR1)

**INSTITUTE OF MARINE AND ENVIRONMENTAL SCIENCES**
UNIVERSITY OF SZCZECIN

Biogeosciences Editorial Board
Copernicus Publications
Bahnhofsallee 1e
37081 Göttingen
Germany

Szczecin, March 10, 2025

Dear Dr. Lever,

We thank you for the opportunity to revise our manuscript entitled 'Controls on Palaeogene deep-sea diatom-bearing sediment deposition and comparison with shallow marine environments'. We have added minor corrections to typos, marked-up in the manuscript, and corrected figure 3, where the legend had been swapped between the Pacific and Atlantic oceans. Please find below detailed responses to the reviewer's suggestions. We have also clarified what makes our paper a novelty, in accordance with your request.

Yours sincerely,
Cécile Figus, on behalf of all Co-Authors

[Figure]

**Responses to Eunah Han:**

General Comments:
This paper offers a paleo-oceanographic insight into the importance of nutrient availability and ocean circulations versus climate on the abundance of diatoms in water columns. This study's wide spatial and temporal scope broadens its implication, encompassing four of the five oceans from the late Mesozoic to the Cenozoic periods. Despite this strength, I am not convinced about the novelty of this paper . Rather, I see it in terms of a summary review rather than an original research article. This is because most of the key data in this paper is pulled from previous literature. Most newly displayed datasets in this paper are analytically rudimentary based on enumeration (mostly the number or frequency of diatom-bearing sediments). This means that insufficient evidence has been presented to support new findings as it stands.

The novelty of this paper lies in the compilation of published local-scale (sites) results, and in the comparison of deep-sea diatom-bearing sediments with our previous study on shallow marine diatomites. Although the data we use are published, no global compilation of deep-sea diatom-bearing sediments has been produced before, and the compilation of these data presents new scientific results, which are not visible without all the data being brought together. Previous studies compiling deep-sea data are on a smaller scale, such as that of Witkowski et al. (2021), so the development of a global-scale study demonstrates differences in diatom accumulation from ocean to ocean. We have added a few sentences to the text to clarify the innovative aspects of our article, for a better understanding by the reader.

Section 3.2. in the Results and Interpretation section is the core part of this paper, however, it has been augmented with a summary of the previous findings. The author's tone fails to convincingly convey the proposed new ideas (e.g., words such as "probably" , "might" , or "may" are used when stating new insights). The writing style of this paper is better suited to a technical report rather than a journal paper, focusing on the description of data or findings instead of the critical assessment or the logical connection between them. Therefore, I recommend the authors position this manuscript as being a foundation for another original study, instead of being an independent journal publication. Alternatively, this paper can be condensed into a one-page "Perspective" article, assuming that the main premise behind this study is critical and timely for the current paleo-sciencific readership.

The discussion is not augmented by a summary of our previous study but, as stated in the title, the second objective of this article is to compare these deep-sea results with those from the shallow marine environment, due to the different diatom responses in these two environments during the diatomite gap (middle Eocene). It is therefore necessary to present our previously published data on shallow marine diatomites in order to highlight the differences in results and controlling factors.

Specific comments:
1. L9-10: Please clarify if the diatomite gap was observed in shallow marine or deep-sea sediments. The diatomite gap typically refers to the deposition in shallow marine sediments. This is essential to follow your logical reasoning that the tectonic reorganizations led to the diatomite gap, despite the deep-sea diatom deposition.

The diatomite gap was indeed observed in shallow marine sediments.

2. L14-28: This part lacks any plausible explanation about how tectonics could have affected the deep-sea diatom deposition, although it does describe a potential impact of the other two factors, climate and ocean circulations.

We have added a part to the introduction to clarify this issue: 'The supply of $H_4SiO_4$ is closely linked to the weathering of continental rocks, which can increase as a result of an intensification of the hydrological cycle, tectonic activity or an increase in atmospheric $CO_2$, for example (Berner et al., 1983; Penman et al., 2019).'

3.  I cannot see any sign that the sampling bias correction method in this study has been significantly advanced, nor that it is novel when compared to the existing statistical methodology.

This method is commonly used to correct sampling bias in palaeobiology. We studied the possibility of other biases, but these are negligible in the context of this article.

a. Section 3.1 Impact of Biases on the Observed Pattern in the Results and Interpretation section may not merit its own independent subsection.

Although this study may still be subject to other biases, not to acknowledge the main biases in our dataset would have a significant impact on the interpretation of the results. Furthermore, sediment preservation and chertification are topics of interest in studies of early Cenozoic sediments.

b. Figure 3 could be included as supplementary information, not as a main figure.

We would like to thank the reviewer for their comment, but we do not agree with it.

4. This paper fails to (i) succinctly introduce a scientific principle or known facts and (ii) connect it to support the original dataset. The latter part (ii) is much more important than the former (i) for a research article. Also, I note that some sentences are logically incomplete. For example, in L145-149:
a. In general, N2-fixing microbes and diatoms occupy distinctive spatial or temporal niches, oligotrophic and warm waters vs. nutrient-rich cold waters. The statement in L145-149 reads as if N2 fixers and diatoms dwell close to each other and exchange nutrients. This issue can be partially solved by elaborating on "then nitrite (NO2-) and nitrate (NO3-) by nitrification" in a new separate sentence. Were the study areas unaffected by other N sources such as terrestrial input and nitrate upwelling? You may want to add more references about known and potential N sources of the time span and specific sites, in addition to Kast et al. (2019).

This has been corrected.

b. Please clarify whether you are referring to benthic diatoms or planktonic diatoms. Their morphology, species, and the d15N values all differ.

Marine sediments contain both planktonic and benthic forms.

c. The sentence in L147-149 can be made clearer by stating what phytoplankton vs. bacterial decomposers do in N cycling, rather than distinguishing phytoplankton vs. diatoms and other organisms. "Used" and "recycled" does not sound mutually exclusive, and "other organisms" is ambiguous.

This has been corrected.

d. Please add a sentence about what proxies you analyzed or cited for N cycling, as you did for the Sr and Li isotopes. You can inform (i) how the analysis is related to diatom accumulation (e.g., lower/higher d15N of sediments suggests more/less diatom deposition) and (ii) if you expected N dynamics to be related to pCO2, as Sr, Li, silicate, and P all are.

We have added a part to section 3.2 to develop the links between diatoms, ocean stratification and d15N.

5. The authors state the aim of this study is to identify the factors determining the vertical flux of diatom deposition. In my opinion, the actual goal should be to advance learning in paleoenvironmental conditions, such as nutrient distribution, ocean circulations, plate tectonics, or climate, based on the diatomite data. Thus, I would expect a revised manuscript to discuss the broader implications of the paleoenvironment.

We would like to thank the reviewer for their comment, but we think there is a misunderstanding. Throughout the article, our data provide information about palaeoenvironments.

Technical corrections:
1. Please be more direct and specific at the individual sentence level. For example:
a. The Abstract section overuses vague expressions such as "response to", "is mainly controlled by", "an indirect correlation", "linked to", and "a comparison of". Clearer, on-point wording, such as increase/enhance/elevate/intensify or decrease/suppress/prevent/weaken, can be substituted for some of these ambiguous expressions. Adding an adverb or an adjective would help inform the scale of change, such as largely, slightly, or x-fold.

Expressions considered vague in this comment cannot be replaced by words such as increase/ enhance/etc, as the subject treated in this paper covers a large dataset over a long period of time. Our results are therefore more diverse than e.g. a trend of constant increase, and deep-sea diatom-bearing sediment deposition responds to several controls that have a different impact depending on the combination of factors present at different periods.

b. L33-34: The summary of Figus et al. (2024a) should be more direct. How did climatic and tectonic factors and ocean circulations change diatomite accumulation? Please specify the factors and the changing direction (e.g., increase or decrease).

As with the previous comment (1.a.), the results obtained in Figus et al. (2024a) are not binary. Climate, tectonic activity and ocean circulation had a more complex impact on shallow marine diatomite deposition than a simple increase or decrease throughout the Palaeogene. Explaining the details of these controls on shallow marine diatomite deposition would make the introduction too

long. Furthermore, as the second objective of this article is to compare the results of the present study with Figus et al. (2024a), the factors and changing directions are to be discussed in section 3 Results and Interpretation.

c.   The writing from Section 3.2.1 onwards is clearer than the earlier part.
2.   L26: "more" seems unnecessary.

'More' has been removed.

3.   L25-47: Please break this paragraph up into two or three paragraphs.

This paragraph is now divided into three paragraphs.

4.   L28-31: The second sentence is redundant in terms of the point made in the preceding one. You can combine these two sentences together and add Brylka et al. (2024) to support the limitation of the datasets.

We have modified this part to remove redundancy: 'Despite several studies examining temporal trends in early Cenozoic diatom-bearing sediments from various locations (Barron et al., 2015; Wade et al., 2020; Witkowski et al., 2021), the global distribution of deep-sea diatom-bearing sediments and the involvement of diatoms in biogeochemical cycling in the Palaeogene oceans tend to be based on very limited datasets (Bryłka et al., 2024).'

5.   L28-30: Did the "several studies" investigate different parts of the oceans or the same location? Please specify which it was in this sentence.

'From various locations' has been added to this sentence to clarify this point.

6.   L34-35: "the presence of a gap" can be just "a gap".

'The presence of' has been removed.

7.   L140: The opening sentence can be more generic, summarizing your approach rather than describing a figure. You can amend this sentence accordingly: "We analyzed x (e.g., four) geochemical proxies to reconstruct the distribution of diatoms (Fig. 4)." You cannot use this sentence if those proxies were analyzed for other papers, rather than for this one.

The proxies mentioned in this paragraph were not analysed as part of this study.

8.   Please make sure that the y-axis of Figure 2(b) covers the entire range of the line graph. The current axis does not encompass a range of higher values greater than 2 ‰.

This has been corrected.

9.   Please add the unit of the y-axes in the following figures: The first two y-axes in Fig. 1; Fig. 2b; Fig. 4a.

Figure 2b has been corrected. Figure 1 (now 7 with the modifications made to the article) shows the number of diatom-bearing sediments, the number of cherts and the number of diatom species. There are no units to add to this figure. Figure 4a (now 8a) is a ratio, there is no unit.

10. You can add a map of sampling sites with sampling frequency if you want to include some graphic information about the geographic coverage of the sampling sites.

A map has been added to the text of the article.

[Figure]

**Responses to Anonymous Referee #2:**

I went through the manuscript titled "Controls on Palaeogene deep-sea diatom-bearing sediment deposition and comparison with shallow marine environments", submitted by Figus and co-authors, to be considered for publication.

The manuscript addresses a very relevant issue, i.e. the relationships between pelagic and shallow marine diatomaceous deposition and the climatic, oceanographic and tectonic reconfigurations occurred during the Palaeogene. A noteworthy task, approached through the review of an impressive amount of data from DSDP, ODP and IODP oceanographic campaigns. The Palaeogene diatomaceous deposition is poorly studied if compared to the Neogene one, and this makes authors' efforts even more appreciable. Works like this are absolutely needed to fix ideas, highlighting the controversies and raising new research questions. Even if the 'smoking gun' is still difficult to be found, this is already an important step beyond. Therefore, on my opinion this manuscript definitely deserves to be published.

Below I provide some suggestions and I raise some questions mostly aimed at improve the discussion. I am very looking forward to know authors' opinion about.

General Comments:

Proxies (isotopes etc.) should be introduced in Materials and methods, while they suddenly appear at lines 140-145. I suggest to include a new paragraph where proxies are presented, and their significance for this investigation briefly described.

We have moved the opening paragraph from section 3.2 Impact of nutrients and ocean circulation to section 2.4 Comparisons with other proxies, in order to introduce proxies in the Material and Methods and to avoid redundancy between the two sections.

I believe that in the Discussions the authors should consider, besides the work of Muttoni and Kent (2007), the very recent results presented by Varkouhi et al. (2024) in a paper titled 'Pervasive accumulations of chert in the Equatorial Pacific during the early Eocene climatic optimum' (Marine and Petroleum Geology 167, 106940). Another relevant work to be considered is Cermeño (2016) - The geological story of marine diatoms and the last generation of fossil fuels (Perspectives in Phycology 3-2, 53-60).

We would like to thank the reviewer for these suggestions, but the comparison of our results with those of Muttoni and Kent (2007) is only intended to highlight the issue of sediment preservation. We do not believe that the work of Varkouhi et al. (2024), although an interesting article, is suitable in the context of our study. However, we have included Cermeño (2016) in the text.

On my opinion, the role of tectonics is rather vague, and must be better addressed. Among others, I suggest to consider the paper of Rea et al. (1990) - Global change at the Paleocene-Eocene boundary: climatic and evolutionary consequences of tectonic events (Palaeo3 79, 1-2, 117-128).

It occurred to us that the way in which we presented the role of tectonics throughout the article was misleading, giving the reader the impression that the impact of tectonics is greater than it actually is. We have therefore slightly modified the text to clarify the indirect role of tectonic activity on the distribution of deep-sea diatom-bearing sediments.

[Figure]

Did you consider the possible role of sea-level changes occurred during the Palaeogene in governing the shallow marine vs deep-sea diatomaceous deposition?

We did consider sea level fluctuations in both studies, but the sea level does not appear to influence diatom-bearing sediment deposition in the deep-sea.

Although I know that this may go a bit far beyond the aims of the present paper, I believe that the discussion might benefit from a brief evaluation of the ecological meaning of the major diatom taxa identified in deep-sea Palaeogene sediments. Two of the authors provided excellent contributions about (see Renaudie et al., 2010 - Siliceous phytoplankton response to a Middle Eocene warming event recorded in the tropical Atlantic (Demerara Rise, ODP Site 1260A). Palaeo3 286, 3-4, 121-134; Renaudie et al., 2018 - The Paleocene record of marine diatoms in deep-sea sediments. Fossil Record 21, 183-205; Witkowski et al., 2020 - Early Paleogene biosiliceous sedimentation in the Atlantic Ocean: Testing the inorganic origin hypothesis for Paleocene and Eocene chert and porcellanite. Palaeo3 556, 109896, etc.). In these works, the widespread occurrence of representatives of the order Hemiaulales is reported. Authors are surely aware that this order comprises diatoms often involved in diatom-diazotroph associations (DDAs) in stratified, oligotrophic waters. Don't authors think that would be interesting to deepen this aspect, especially in the light of the variations of nitrogen isotopes (see for example paragraph 3.3. in Knies et al., 2008 - Surface water productivity and paleoceanographic implications in the Cenozoic Arctic Ocean. Palaeocenography and Palaeoclimatology 23, 1)?

This would be an interesting perspective for further study, but it would be a more appropriate subject for a new article. To do this, we would need detailed taxonomic work for each of the sites in the database, which unfortunately is not available. Some site reports only indicate traces of diatoms without any further precision, or give a very brief indication of some of the diatoms identified, and no taxonomic work has subsequently been carried out on these sites. We would therefore have to study samples from sites for which we lack detailed taxonomic descriptions, which is not part of the present study.

What can be inferred from the Palaeogene sedimentary record about the other components of the pelagic trophic chain (not only radiolarians)? Any significant event among other groups of phyto- or zooplankton (or among marine metazoans) that can be relevant for the present study?

During our study, we compared our results with other records of marine planktonic organisms from the NSB Database. No significant events were found to be common to our results and these other records. However, a higher resolution study might reveal links between diatom distribution and other groups of phyto/zooplankton.

Specific comments:
Line 14: 'in the oceans' – 'in the modern oceans'

This sentence has been modified to add 'modern'.

[Figure]

Line 18: 'supplied' – 'mostly supplied'; a reference is needed here, I suggest Tréguer et al. (2021) - Reviews and syntheses: The biogeochemical cycle of silicon in the modern ocean. Biogeosciences 18 (4), 1269-1289.

We have added the word 'mostly' and the reference Tréguer et al. (2021) to this sentence.

Line 22: 'gravitational sinking of diatoms in the water column' – Some references are needed here; moreover, I would expand a bit this passage. The sinking of diatom aggregates is much more than a passive process controlled by gravity. I recommend the seminal works authored by Smetacek and Grimm about this aspect. For example: Smetacek (1985) – Role of sinking in diatom life-history cycles: ecological, evolutionary and geological significance. Marine Biology 84, 239-251; Grimm et al. (1997) – Self-sedimentation of phytoplankton blooms in the geologic record. Sedimentary Geology 110 (3-4), 151-161.

The sentence has been amended and references added.

Line 58: 'opal-CT' – 'opal-CT and quartz'.

We decided not to add 'quartz' to this sentence, to avoid any redundancy with the sentence in line 61.

Line 61: 'opal-CT (siliceous microfossils)' – Unclear, I suggest to rephrase. It seems from this passage that siliceous microfossils are by definition composed of opal-CT.

We have deleted '(siliceous microfossils)' from this sentence to improve clarity.

Line 62: 'diatoms are usually obliterated' – An interesting exception, just for authors' information: Hein et al. (1990) - Eocene diatom chert from Adak Island, Alaska. Journal of Sedimentary Research 60 (2), 250-257. When dealing with the diagenesis of biosiliceous remains, I suggest to cite the processes of reverse weathering (see Michalopoulos & Aller, 1995 – Science 270, 5236, 614-617) and pyritization. This latter can be very interesting for this case study, because it has been described in Palaeogene diatom-bearing sediments of Northern Europe: De Jonghe et al. (2011) - Middle Eocene diatoms from Whitecliff Bay, Isle of Wight, England: stratigraphy and preservation. Proceedings of the Geologists' Association 122, 472-483.

Thank you for sharing this very interesting paper on diatom-rich quartz chert. We have added a sentence to include reverse weathering and pyritisation in the text.

Line 126: 'surface/volume ratio' – References are needed here.

We have added Hein et al. (1990) for the reference.

Line 225: '(…) shallow marine diatomite(s) deposited in open ocean conditions' – A bit confusing…

An explanation has been added to the sentence to clarify this point.

[Figure]

Figures:
It would be nice to see some maps in the maintext, similar to those reported in Figures 1, 4 and 5 in Figus et al. (2024).

We have added 3 figures to show the distribution of diatom-bearing sediments throughout the Palaeogene.

A very simple, resuming sketch (or a flow chart) could greatly help the reader to fix the major points discussed in the text.

We have added a resuming sketch to the article.

---

## Author Response (AR2)

INSTITUTE OF MARINE
AND ENVIRONMENTAL SCIENCES
UNIVERSITY OF SZCZECIN

Biogeosciences Editorial Board
Copernicus Publications
Bahnhofsallee 1e
37081 Göttingen
Germany

Szczecin, April 2, 2025

Dear Dr. Lever,

We thank you for the opportunity to revise our manuscript entitled 'Controls on Palaeogene deep-sea diatom-bearing sediment deposition and comparison with shallow marine environments'. We have made all the corrections requested by the reviewer.

Yours sincerely,
Cécile Figus, on behalf of all Co-Authors

[Figure]

**Responses to Anonymous Referee #2:**

Throughout the maintext, carbon dioxide is often reported as 'CO2' (2 is not subscripted).

This has been corrected.

Line 3: 'sediments occurrences' -> 'sediment occurrence'.

This has been corrected.

Line 10: '(…) during the diatomite gap' -> '(…) during the diatomite gap in shallow water settings'.

This sentence has been modified to add 'in shallow water settings'.

Lines 29-32: I suggest a small, additional modification aimed at improve the passage (but feel free to modify it as you prefer, considering the further reference that I included). 'Another aspect of the biological pump is the "self-sedimentation" of diatoms in the water column (i.e., diatoms seem to have developed mechanisms to accelerate gravitational mass sinking of their skeletons in the water column), leading to the burial of diatom frustules in sediments (Smetacek, 1985; Grimm et al., 1997).' -> 'Another aspect of the biological pump is represented by the processes accelerating the diatom sinking in the water column and the consequent burial of diatom frustules. Among them, the massive settling of large aggregates of bloom-forming or stratified-adapted diatoms ("self sedimentation" and "fall dump", respectively) play a prominent role in enhancing diatom-bearing sediment formation (Smetacek, 1985; Grimm et al., 1997; Kemp et al., 2000).'

The sentence has been amended.

Line 340 – typo: remove 'the' before 'present'.

This has been corrected.

Line 354 – typo: split 'thatassumes'.

This has been corrected.

Figures
Figure 1 – Nice and useful sketch! Just a few suggestions (mere formality…):

I would like to thank you for your comments. I am glad that this new figure has also been revised, as I drew it from scratch.

1) Why do you report the chemical formula of orthosilicic acid while P and N are just reported as chemical symbols? I suggest to simplify and uniform: just "Si", "P" and "N".

This has been corrected.

2) I would remove the term 'self-sedimentation', which is actually only a process among many others; "sedimentation" is enough on my opinion.

This has been corrected.

3) Why upwelling should bring only nitrogen to the surface waters? I would report also Si and P.

This has been corrected.

Figure 3 (caption, line 2): "The agenta and grey circles" -> "The magenta and grey circles".

This has been corrected.